# Maternal Exposure to Ambient Air Pollution and Risk of Preeclampsia: A Population-Based Cohort Study in Scania, Sweden

**DOI:** 10.3390/ijerph17051744

**Published:** 2020-03-07

**Authors:** Yumjirmaa Mandakh, Ralf Rittner, Erin Flanagan, Anna Oudin, Christina Isaxon, Mary Familari, Stefan Rocco Hansson, Ebba Malmqvist

**Affiliations:** 1Division of Occupational and Environmental Medicine, Department of Laboratory Medicine, Lund University, Scheelevägen 8, Building 402A, 22381 Lund, Sweden; yumjirmaa.mandakh@med.lu.se (Y.M.); ralf.rittner@med.lu.se (R.R.); erin.flanagan@med.lu.se (E.F.); anna.oudin@med.lu.se (A.O.); 2Department of Ergonomics and Aerosol Technology, Lund University, Sölvegatan 26, Box 118, 22100 Lund, Sweden; christina.isaxon@design.lth.se; 3School of BioSciences, University of Melbourne, Parkville, Melbourne 3010, Australia; m.familari@unimelb.edu.au; 4Division of Obstetrics and Gynecology, Department of Clinical Sciences, Lund University, Klinikgatan 12, 22185 Lund, Sweden; stefan.hansson@med.lu.se

**Keywords:** Ambient air pollution, preeclampsia, small-for-gestational age, environmental epidemiology

## Abstract

The aim of this study was to investigate the risk of developing preeclampsia (PE) associated with gestational exposure to ambient air pollutants in southern Sweden, a low-exposure area. We used a cohort of 43,688 singleton pregnancies and monthly mean exposure levels of black carbon (BC), local and total particulate matter (PM_2.5_ and PM_10_), and NO_X_ at the maternal residential address estimated by Gaussian dispersion modeling from 2000 to 2009. Analyses were conducted using binary logistic regression. A subtype analysis for small-for-gestational age (SGA) was performed. All analyses were adjusted for obstetrical risk factors and socioeconomic predictors. There were 1286 (2.9%) PE cases in the analysis. An adjusted odds ratio (AOR) of 1.35 with a 95% confidence interval (CI) of 1.11–1.63 was found when comparing the lowest quartile of BC exposure to the highest quartile in the third trimester The AOR for PE associated with each 5 µg/m^3^ increase in locally emitted PM_2.5_ was 2.74 (95% CI: 1.68, 4.47) in the entire pregnancy. Similar patterns were observed for each 5 µg/m^3^ increment in locally emitted PM_10_. In pregnancies complicated by PE with SGA, the corresponding AOR for linear increases in BC was 3.48 (95% CI: 1.67, 7.27). In this low-level setting, maternal exposure to ambient air pollution during gestation was associated with the risk of developing PE. The associations seemed more pronounced in pregnancies with SGA complications, a finding that should be investigated further.

## 1. Introduction

Ambient air pollution is a dynamic mixture of ultrafine, fine and coarse particles as well as various gases. Long-term exposure to ambient particles of less than 2.5 micrometers in diameter (PM_2.5_) was the fifth leading risk factor for global deaths in 2015 [1]. Moreover, 84% of the global population is exposed to annual mean PM_2.5_ levels exceeding the “WHO air quality guidelines -global update 2005” [2]. Little is known about which PM_2.5_ properties, in terms of particle size distribution, chemical constituents, particle shape, and surface reactivity, are responsible for its toxicity. The estimated 4.2 million premature deaths linked to PM_2.5_ were mainly attributed to ischemic heart disease, cerebrovascular disease, chronic obstructive pulmonary disease and lung cancer among adults, as well as lower respiratory infections for children under age five [1].

Adverse health effects following air pollution exposure seem to be especially substantial for vulnerable subgroups of the population, such as pregnant women and their unborn children [3,4]. For instance, exposure to ambient air pollution during pregnancy poses a significantly increased risk of low birth weight and prematurity [5,6], as well as maternal health conditions, such as preeclampsia [7]. Preeclampsia (PE) is a pregnancy-induced hypertensive disorder marked by a new-onset hypertension and one or more maternal manifestations including proteinuria, maternal organ dysfunction, or uteroplacental dysfunction presenting after 20 gestational weeks [8]. This condition is the second main cause of maternal death worldwide and one of the leading causes of adverse fetal outcomes, such as low birth weight and prematurity [9,10,11,12]. The widely used clinical subclassification for assessing the severity of preeclampsia includes early- and late-onset PE: diagnosed at their occurrence before and after 34 weeks of gestation, respectively [8]. 

Though the etiology is still enigmatic, the placenta has been deemed central for the development of preeclampsia [13]. As originally described, PE was considered to be a two-stage disorder that first consisted of a reduced placental perfusion due to a shallow remodeling of spiral arteries, followed by the maternal clinical manifestations of PE [14]. The more recent, modified two-stage model of PE explicitly elaborates on the difference between early- and late-onset PE based on the induction time of placental dysfunction and maternal susceptibility risk factors, such as chronic vascular disease [13]. The model describes several pathophysiologic mechanisms including oxidative stress, inflammation, senescence at the first stage, and general endothelial damage at the second stage [15,16,17]. However, the initial trigger of the disorder is still not known. In pursuit of this, the literature on the development of PE associated with maternal exposure to air pollution during gestation is expanding [7,18].

Several epidemiological studies have reported a positive association between preeclampsia and maternal exposure to gaseous air pollution components, such as ozone (O_3_), nitrogen dioxide (NO_2_), and sulfur dioxide (SO_2_) in both relatively low and high exposed areas [19,20,21,22,23,24]. Furthermore, maternal exposure to particulate matter less than 10 micrometers in diameter (PM_10_) as well as PM_2.5_ during pregnancy has been found to be positively associated with a higher risk of PE [19,24,25,26]. However, some studies have found no such associations [27,28,29]. The inconsistencies in findings can be partly explained by the varying study settings as well as different methods of exposure assessment. It is, therefore, of importance to further study the possible association between PM_10_, PM_2.5_, and black carbon (BC) and risk of PE in a low-level setting, using high-spatial resolution PM models.

## 2. Materials and Methods 

### 2.1. Study Design and Setting

The study was based on a cohort of 43,688 singleton pregnancies during the 2000–2009 period in Scania (Skåne) county in southern Sweden, which has a total population of approximately 1.35 million [30]. A detailed description of the original birth cohort, Maternal Air Pollution in Southern Sweden (MAPSS), has been previously given by Malmqvist et al. [31]. MAPSS consists of a local birth register, Perinatal Revision Syd (PRS), connected to individual exposure assessments of air pollution levels at each maternal residential address within the catchment area of hospitals in Malmö, Lund and Trelleborg (Figure 1). PRS is regarded as a high-quality medical birth register with 98% coverage of all births in Scania. Additionally, this birth register is linked to sociodemographic and socioeconomic factors obtained from Statistics Sweden through each person’s unique personal identification number (PIN) [32]. 

In total, MAPSS consists of 48,777 singleton pregnancies from 1999 to 2009. However, we restricted our study population to women giving birth during the years 2000–2009, as particle exposure data were limited to this period. Thus, individual exposure assessment was completed for 43,688 singleton pregnancies. Figure 2 illustrates the flowchart of this study. After excluding any missing values from categorical variables, the complete case analysis consisted of 35,570 singleton pregnancies, also from 2000 to 2009.

### 2.2. Exposure Assessment

A Gaussian plume air dispersion model was developed to extrapolate the monthly mean air pollution concentrations in Scania, Sweden from 1999 to 2009 [33]. This flat two-dimensional dispersion modeling was constructed using the software program ENVIMAN, a locally adjusted version of the American Meteorological Society/Environmental Protection Agency Regulatory Model (AERMOD) by the United States Environmental Protection Agency (USEPA) [34]. The dispersion model is considered to be a more sophisticated approach to estimating the spatiotemporal variability of locally emitted traffic-related air pollution as compared to other models, such as land-use regression models [18]. Two years were modeled for PM and, consequently, its BC component: 2000 and 2011 at 100 m by 100 m spatial resolution; the interpolation of the years and months between them was based on an atmospheric ventilation index using year- and month-specific meteorological parameters [33]. Air pollution data for nitrogen oxides (NO_X_), modeled for each year, were derived from two databases: modeling for the years 1999–2005 used 500 m x 500 m spatial resolution, and the years 2006 to 2009 were paired with 100 m grid cells. Again, NO_X_ concentration values for the year 1999 were excluded.

Eight categories of emission sources were included: road traffic, shipping, aviation, railroads, industries and major energy and heat producers, small-scale heating, non-road vehicles, and local emissions in Zealand, Denmark. The Swedish Road Administration’s data detailing types of vehicles, fuel sources, and speed limits, among others, were used to account for emissions from road traffic. Shipping emissions for the year 2000 were estimated by Gustafsson [34], and 2011 emissions were described by Project Shipair [35]. Aviation emission data were obtained from Scandinavian airports’ annual environmental reports. Because railroads in Sweden are mainly electric, railroad emissions were estimated using the fuel consumption of the few operational diesel engine freight trains both on railway lines and at railway stations [34]. 

The reported emissions from industry and energy production were also included. Regarding small-scale heating, chimney sweeping records from the National Rescue Agency were used to estimate the frequency of stove use [34]. Further, the emissions from non-road vehicles were obtained from a report by the Swedish Environmental Research Institute, IVL. Due to Scania’s proximity to the industrial island Zealand, Denmark, their local emissions were incorporated in the dispersion modeling [33]. 

To estimate the levels of BC, PM_2.5_, PM_10_, and NO_X_ from the aforementioned local emission sources, the emission factors developed by HBEFA 3.2 and Project Transphorm were used for the proportional distribution of emission sources. Moreover, the temporal and spatial variations of air pollution were incorporated with the atmospheric ventilation index through a complex method developed by the SCAC project [33]. The hourly levels of particles were estimated through linear interpolation in the model and then aggregated into monthly mean levels, approximately corresponding to the calendar months of the pregnancy period.

PM_2.5_ and PM_10_ have particularly long atmospheric lifetimes and can, therefore, be transported and deposited over long distances and contribute to the ambient air pollution levels in Scania. The dispersion modeling of these non-local particles was captured using PM_2.5_ and PM_10_ levels measured at meteorological background stations [33]. Thus, total PM_2.5_ and PM_10_, and their subsequent BC components, were calculated as the sum of locally emitted PM_2.5_ and PM_10_ and the long-range transport of background PM_2.5_ and PM_10_ deposited in Scania, Sweden over the 2000-2009 period. Total NO_X_ concentrations consisted of the sum of local NO_X_ levels together with 2.5 µg/m^3^, the mean NO_X_ exposure level derived from a rural background site.

### 2.3. Variables

#### 2.3.1. Outcome Variable

The outcome variable in this study was preeclampsia, specifically all preeclampsia cases (total PE) regardless of severity, which was calculated by combining both moderate and severe PE cases recorded in MAPSS. In the Swedish Medical Birth Register, preeclampsia, excluding superimposed PE, is defined and diagnosed in accordance with the Swedish adaptation of the 10th version of the International Statistical Classification of Diseases and Related Health Problems (ICD-10) by the World Health Organization (WHO) [36]. 

#### 2.3.2. Exposure Variable

We obtained the geographical coordinates of the residential address of each woman from Statistics Sweden and used these data to calculate her individual exposure. As this address database is only updated annually, any changes of residency can only be updated at the end of the calendar year. Therefore, we estimated exposure for every gestational month based on the nearest available time: January–June coordinates from the end of the previous year and July-December coordinates from the end of the current year.

The atmospheric aerosol particles examined in this study were locally emitted black carbon, PM_2.5_ and PM_10_ as well as total PM_2.5_ and PM_10_, which includes regional background levels (i.e., those generated elsewhere that have traveled into the area) in addition to local emissions. BC, a component of PM, is a mixture of carbonaceous substances in atmospheric aerosol formed from incomplete combustion processes [37]. We have also included analyses for NO_X_; this is because we have investigated PE and NO_X_ in a previous study [20] but can only now adjust for socioeconomic status (SES) variables. However, the main emphasis of this study is on atmospheric particles (BC and PM variations).

The MAPSS database consists of monthly mean levels of BC, local and total PM_2.5_ and PM_10_ from 2000 to 2009 as well as NO_X_ from 1999–2009. To ensure consistency, NO_X_ exposure data from 1999 were excluded, and all analyses were limited to the 2000–2009 period. For this spatiotemporal analysis, we aggregated the monthly mean levels into four exposure windows representing different gestational periods: the individual trimesters and the entire pregnancy period. As the diagnosis date of PE is not available in the Swedish Medical Birth Register, we used gestational age (GA) at delivery to calculate trimester exposure levels. GA was based on the first ultrasound measurement performed in the first trimester (around 14 gestational weeks) [38]. 

#### 2.3.3. Predictors

The obstetrical risk factors connected to MAPSS and included in this study were maternal country of origin (Nordic-born or other countries), maternal age (≤19, 20–34, ≥35), parity, pre-pregnancy body mass index (BMI, <18.5, 18.5–24.9, 25–29.9, ≥30, missing), smoking habits at first antenatal visit (non-smoker, 1–9 cigarettes/day, ≥10 cigarettes/day, missing), diabetes mellitus, gestational diabetes, essential hypertension, and gestational hypertension (yes or no) as well as fetal sex, year, and season of delivery. The MAPSS database also contains information on socioeconomic predictors; those considered were maternal education level (pre-secondary, secondary, and post-secondary) and annual household income (<200,000, 200,000–300,000, 300,000–400,000, >400,000 SEK/year). Maternal country of birth is also considered an indicator for one’s socioeconomic status here.

We selected potential confounders based on a priori knowledge. These included maternal country of origin, young (<20 years) or old (>35 years) maternal age, smoking, low or high body mass index, low socioeconomic status, and gestational diabetes [4,20,39,40,41,42,43]. To further understand potential confounding factors, we used a directed acyclic graph (DAG) [44] (Appendix A). In this graph, the covariates are illustrated as nodes and are connected to each other by directed paths representing causal links. In addition to the aforementioned confounders, the year and the season of delivery were identified as residual confounders in previous literature [7]. These are depicted as a common cause (red nodes), representing a confounding effect on the causal path of DAGs (Appendix A). The blue nodes are the ancestors, including parity, diabetes mellitus, gestational diabetes, essential hypertension, gestational hypertension, and fetal sex. These identified confounders were applied to our logistic regression models. 

### 2.4. Statistical Methods

Binary logistic regression models were applied in univariate analyses and multivariable analyses after adjusting for the effects of maternal age, parity, maternal pre-pregnancy body mass index, maternal smoking, and the year of birth (2000–2009). 

We performed logistic regression analyses for all pollutants over the entire period of pregnancy as well as at each trimester separately. Correlations between the exposure variables were significantly high, especially the locally emitted particles, resulting in Pearson correlation coefficients around 0.9 (Appendix A). Therefore, only single-pollutant models were used. 

The main analysis, a complete case analysis (CCA), excluded all observations with missing values in the categorical predictors. In addition, gestational diabetes, gestational hypertension, and essential hypertension were identified as intermediate variables based on the DAG. Therefore, we conducted our main analysis with or without these intermediate variables. Here, linear exposure trends were assessed for atmospheric particles only, using a continuous increment of a 1 µg/m^3^ increase in BC levels and a 5 µg/m^3^ increase in local and total PM_2.5_ and PM_10_ levels. An additional CCA was performed using exposure-specific quartiles for all pollutants under investigation. As a supplementary investigation, we also ran a logistic regression analysis for exposure-specific quartiles of all pollutants with missing values included for late-onset preeclampsia, a clinically relevant subtype of PE. Finally, a sensitivity analysis considering PE with or without a small-for-gestational age (SGA) fetus was conducted considering both quartile-specific exposure as well as linear exposure increases, which was limited to the ambient particles only. SGA is defined as having a birth weight less than the 10th percentile.

All statistical analyses were carried out using IBM SPSS Statistics version 25. Adjusted odds ratios with 95% confidence intervals (CI) were reported for the CCA as main results as well as for the supplementary and sensitivity analyses.

### 2.5. Ethics Approval

The Lund University Ethical Committee approved this study prior to its realization (permission number 2014/696). 

## 3. Results

Out of the sample population of 35,570 births, 1034 (2.9%) of the pregnant women developed PE in Scania during the study period. Preeclampsia was more common among women with high BMI, gestational diabetes, or hypertension (both essential and gestational). PE also developed more frequently if the baby was the first-born or male child. Further, Nordic-born women experienced a higher proportion of PE cases than those born in other countries. Generally, women with PE were less frequent smokers. The PE rates also varied slightly between years with no clear pattern, although PE was more common during summer months. Additional characteristics examined were similar overall between women with PE and the healthy controls (Table 1).

Younger women and women with low SES, considered to be those with low educational attainment, foreign country of birth and low income, were more likely to be living in the highest exposed areas (Appendix A). This exposure pattern was also evident for women with gestational diabetes (Appendix A). Moreover, PE was more prevalent in the highest quartiles of locally emitted BC, PM_2.5_, and PM_10_ (Appendix A).

Exposure levels in Scania over the 10-year period also varied year to year and between seasons (Table 2). During the study period, the annual mean levels of total PM_2.5_ (11.09 ± 1.16 µg/m^3^), PM_10_ (15.81 ± 2.35 µg/m^3^), and NO_X_ (14.82 ± 7.62) were lower than EU guideline values (25 µg/m^3^ for PM_2.5_, 40 µg/m^3^ for PM_10_, and 40 µg/m^3^ for NO_X_) [45]. Examples of local BC, PM_2.5_, and PM_10_ concentration levels for Scania, Sweden over a typical study year are presented in Appendix A, respectively.

Results obtained from the analysis using exposure-specific quartiles suggest an association between exposure to BC and the risk of developing PE (Table 3). For example, an adjusted odds ratio (AOR) of 1.35 with a 95% confidence interval (CI) of 1.11–1.63 was found when comparing the lowest quartile of BC exposure to the highest quartile in the third trimester (Table 3). Results were very similar among the trimesters, and similar effects were also seen for NO_x_ and locally emitted levels of PM_2.5_ and PM_10_ (Table 3). For total PM_2.5_ and PM_10_, an effect of the exposures during the entire pregnancy period was found; however, this was only statistically significant in some trimester-specific exposure windows (Table 3). Appendix A illustrates these effect estimates in graphic form. Without adjusting for gestational diabetes, gestational hypertension, and chronic hypertension (variables that could lie in the causal pathway between air pollution exposure and preeclampsia), results did not substantially change (Table 4). 

Using a linear exposure trend, consistent air pollution effects for the risk of developing PE are shown in Appendix A. For example, the risk of PE significantly increased with each increment of 1 µg/m^3^ in BC (AOR 2.14 (95% CI: 1.48, 3.09)) during the entire pregnancy (Appendix A). It should be stated that the number of women exposed to levels above 1 µg/m^3^ of BC is low, counting 288 (0.6%) cases in the CCA. For each 5 µg/m^3^ incremental increase in other exposure pollutants during the entire pregnancy period, the AORs for PE development were 1.40 (95% CI: 1.13, 1.72) for total PM_10_, 1.98 (95% CI: 1.27, 3.09) for total PM_2.5_, 1.50 (95% CI: 1.19, 1.89) for local PM_10_, and 2.74 (95% CI: 1.68, 4.47) for local PM_2.5_ (Appendix A). Results did not change substantially if not adjusting for the following variables that could be in the causal pathway: gestational diabetes, gestational hypertension, and chronic hypertension, as illustrated in Appendix A. Appendix A illustrates these effect estimates in graphic form. 

In the supplementary analysis of preeclampsia clinical subgroups, early- and late-onset PE, without excluding cases with missing values (i.e., non-CCA), a statistically significant and positive association was found between exposure to locally emitted PM_2.5_ and the risk of late-onset PE. For instance, AORs for late-onset PE were 1.23 (95% CI: 1.02, 1.48) for the third trimester and 1.31 (95% CI: 1.08, 1.58) for the entire pregnancy period (Appendix A). These significant increased risks for late-onset PE continued for local PM_10_ exposure, with AORs of 1.22 (95% CI: 1.01, 1.48) in the second trimester and 1.25 (95% CI: 1.03, 1.51) in the third trimester (Appendix A). A descriptive table of the participants’ characteristics by these clinical subgroups is illustrated in Appendix A. Furthermore, our sensitivity analysis appeared to demonstrate more prominent effect estimates in preeclamptic pregnancies complicated by SGA, indicative of a more severe form of PE (Appendix A). 

## 4. Discussion

The results of this study demonstrate a significant association between maternal exposure to ambient air pollution during gestation and the risk of developing PE, especially for preeclamptic pregnancies complicated by SGA. 

In accordance with the present results, previous studies have demonstrated a strong association between PE and exposure to fine particles at various windows of exposure. Dadvand et al. [25], for example, observed a statistically significant positive association between first trimester PM_2.5_ exposure and early-onset PE as well as third trimester PM_2.5_ exposure and late-onset PE. In a meta-analysis of studies, pooled estimates indicated that each additional increment of 5 µg/m^3^ in PM_2.5_ led to a 31% increased risk of PE [7]. Additionally, positive and statistically significant associations were found between the risk of PE and exposure to PM_10_ from brake dust and combined traffic-related sources [26]. Analyses accounting for the spatial and temporal variations in emission sources as well as the transportation of air pollutants are emerging, allowing for studies on critical windows of air pollution exposure in relation to PE. Still, little is known about the possible associations between maternal exposure to BC and PM components during the different stages of pregnancy and the risk of PE.

With relatively low particle concentrations producing an estimated large magnitude of effect, maternal exposure to air pollution seems to affect both early- and late-onset PE as well as PE with and without SGA. The underlying mechanisms, though not yet entirely understood, could be in line with the oxidative stress and placental senescence processes described in the modified two-stage model of preeclampsia development [13,46]. Due to early placental dysfunction, for instance, early-onset PE is often accompanied by fetal growth restriction (FGR) [13]. Late-onset PE, on the other hand, is induced by prolonged oxidative stress, senescence and hypoxia in a normally developed placenta, thus FGR is rarely seen [13]. It seems possible, then, that air pollution may have different toxic effects depending on the time of exposure. To investigate this possibility, we used proxy indicators for early-onset PE (PE with SGA) and late-onset PE (PE without SGA) based on findings by Redman et al. [13]. In PE cases with SGA, our results suggest that the toxic effects of ambient particles may begin as early as the first trimester, which could effectively induce the first stage of PE development, i.e., the remodeling of the spiral arteries causing insufficient blood perfusion and placental oxidative stress [47]. Similar pathophysiologic mechanisms were exhibited in an animal model when pregnant mice were exposed to PM_2.5_ from diesel exhaust [48]. The BC and local PM_2.5_ exposure in early pregnancy could, therefore, interfere with placental development. If continued throughout pregnancy, these exposures may also affect placental function and aggravate cellular senescence, which could then explain the increased risk of PE complicated by SGA found at all exposure windows in this study.

The toxic effects of air pollution particles may also affect the placenta at the epigenetic level. In a mixed-effects model study on human placentae, for example, placental circadian pathway methylation was found to be significantly and positively associated with third trimester PM_2.5_ exposure [49]. This could possibly be the underlying mechanism of our findings concerning air pollution effects on PE without SGA. For instance, the failed remodeling of spiral arteries causing reduced utero-placental blood flow and subsequent oxidative stress and placental damage from late-onset PE may support the presence of first-stage placental dysfunction. Moreover, methylation of placental DNA, *ADORA2*, has been associated with maternal exposure to NO_2_ during the second trimester and the development of PE [50]. Furthermore, a recent meta-analysis has predicted a genetic predisposition to reduced DNA methylation in preeclampsia associated with air pollution exposure: *methylenetetrahydrofolate reductase* (*MTHFR*) *C677T* gene polymorphisms [51]. This process may, therefore, also trigger the first stage of the two-stage model [52,53,54]. Similar dysfunction was demonstrated in an experimental study on pregnant mice exposed to PM_2.5_ derived from diesel exhaust [48]; however, it has yet to be seen in human pregnancies.

A key strength of this study is the large-sized birth cohort with thorough maternal exposure estimates using high-resolution dispersion modeling of BC, PM_2.5_, PM_10_, and NO_X_ in Scania, Sweden. All pollutants were modeled at a fine time scale (hourly) and with a 100 m by 100 m spatial resolution, with the exception of NO_X_ for the 2000–2005 period, during which 500 m grids were utilized. The total mass of BC, PM_2.5_, PM_10_, and NO_X_ was estimated using the detailed emissions database’s data on over 25,000 exposure points from eight common local sources combined with background levels. Obtaining the geographic coordinates of MAPSS women’s home residences from Statistics Sweden through their PIN helped enhance our exposure assessment’s validity and precision. We also utilized a number of individual-level, SES-related variables from a reliable, national register to control for confounders. Furthermore, the only well-established environmental factor associated with increased incidence of PE to date is living at high altitude [55]. Above 2700 m, an increased expression of the angiotensin type I receptor in the placental renin-angiotensin system causes vasoconstriction and contributes to oxidative stress in the placenta [56,57]. Thus, our study results enhance support for and evidence of ambient air pollution being an additional environmental factor important to the development of preeclampsia. The analysis of BC undertaken here has also addressed the previous uncertainty about its health effects [58] and extended our knowledge of it being a toxic component of both PM_2.5_ and PM_10_. 

Despite several strengths, this study does have certain limitations. As in most epidemiological studies on air pollution, we assessed exposure as outdoor exposure at maternal residency, not accounting for commuting, indoor, or occupational exposure, while the residential data were completely known from a reliable source. Furthermore, the date of PE diagnosis was not available in the Swedish Medical Birth Register. Thus, the precise length of women’s exposure to these time-varying ambient air pollutants prior the development of PE was not possible to determine. This is a common bias reported in other similar studies [18]. Given the uncertainty in the timing of disease presentation, we assumed that the exposure preceded the outcome by considering the gestational date of delivery as the timing of the outcome variable. More specifically, given the fact that the only cure for PE to date is delivery [13], we estimated the length of exposure by the gestational date of delivery, since it is a reliable datapoint recorded in birth registries and is, therefore, widely used in similar studies. Pregnant women exposed at lower levels but not having severe enough PE to present symptoms could also be confounding the ability to identify an association. However, sensitivity analysis in terms of moderate vs. severe PE was not performed, as it is stated that such subclassifications are not clinically relevant due to the rapid and unpredictable worsening of PE [8]. The study is limited by the lack of information on potential confounders, with more than 5% of women lacking data for variables such as smoking and pre-pregnancy BMI. Complete case analysis excludes observations with missing values and may cause bias away from the null. Moreover, when interpreting our results from the linear trends, it should be considered that few women are exposed to levels above the increment, such as 1 µg/m^3^ for BC. 

More research on this topic needs to be undertaken before the association between particle exposure and PE can be completely understood. Also, more information on residual confounding factors such as maternal occupation, physical activity, and environmental tobacco smoke would help to establish a greater degree of accuracy on this relationship. 

## 5. Conclusions

The pregnant women’s exposure levels were generally lower than the limit values of the EU’s current air quality directives [45]. However, both total and local PM_2.5_ exposure levels were above the limit values of the WHO air quality guidelines and the Swedish environmental objective Clean Air. 

The results suggest that maternal exposure to ambient air pollutants during gestation is an important factor that may contribute to the development of PE and SGA. Furthermore, the increased risk of PE associated with the linear exposure trend of ambient particles indicates the importance of reducing PM levels, even in low-exposure areas.

## Figures and Tables

**Figure 1 ijerph-17-01744-f001:**
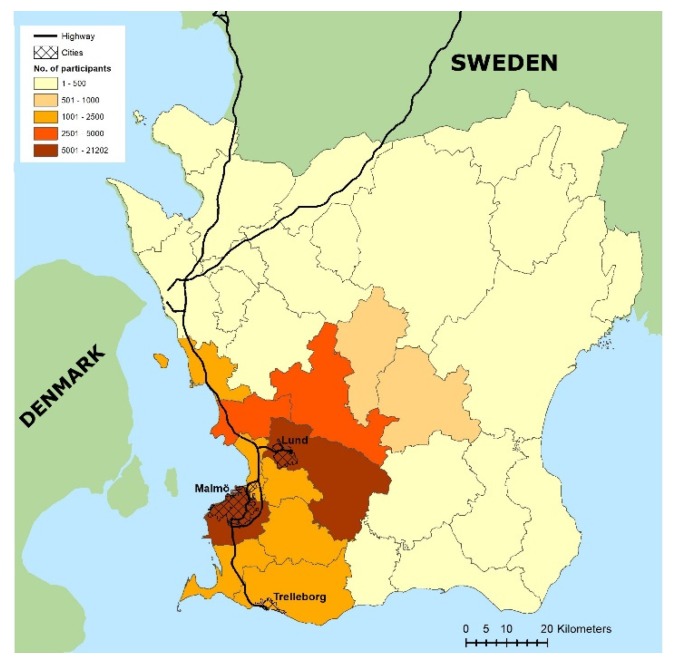
Distribution of the study population of Maternal Air Pollution in Southern Sweden (MAPSS), 1999–2009. Map by Emilie Stroh.

**Figure 2 ijerph-17-01744-f002:**
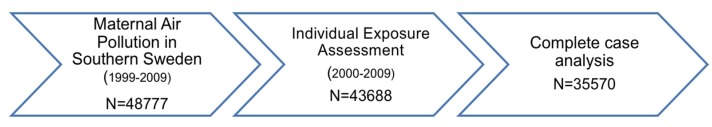
Flowchart of the study.

**Table 1 ijerph-17-01744-t001:** The characteristics of the participants in Scania, Sweden over the 2000–2009 period.

Characteristics	Non-PE, *n* (%)	PE, *n* (%)	Total, *n* (%)
Births			
	34,536 (97.1)	1034 (2.9)	35,570 (100)
Maternal age			
≤19	434 (1.3)	17 (1.6)	451 (1.3)
20–34	27,508 (79.7)	800 (77.4)	28,308 (79.6)
≥35	6594 (19.1)	217 (21.0)	6811 (19.1)
Parity			
Nulliparous	16,464 (47.7)	719 (69.5)	17,183 (48.3)
Parous	18,072 (52.3)	315 (30.5)	18,387 (51.7)
Pre-pregnancy BMI			
<18.5	899 (2.6)	14 (1.4)	913 (2.6)
18.5–24.9	21,801 (63.1)	479 (46.3)	22,280 (62.6)
25–29.9	8410 (24.4)	308 (29.8)	8718 (24.5)
≥30	3426 (9.9)	233 (22.5)	3659 (10.3)
Smoking (cigarettes per day)			
Non-smoker	31,097 (90.0)	964 (93.2)	32,061 (90.1)
<10	2441 (7.1)	48 (4.6)	2489 (7.0)
≥10	998 (2.9)	22 (2.1)	1020 (2.9)
Diabetes Mellitus			
No	34,317 (99.4)	1016 (98.3)	35,333 (99.3)
Yes	219 (0.6)	18 (1.7)	237 (0.7)
Gestational Mellitus			
No	33,647 (97.4)	975 (94.3)	34,622 (97.3)
Yes	889 (2.6)	59 (5.7)	948 (2.7)
Essential hypertension			
No	34,395 (99.6)	1003 (97.0)	35,398 (99.5)
Yes	141 (0.4)	31 (3.0)	172 (0.5)
Gestational hypertension			
No	34,072 (98.7)	903 (87.3)	34,975 (98.3)
Yes	464 (1.3)	131 (12.7)	595 (1.7)
Maternal education			
Pre-secondary	4500 (13.0)	123 (11.9)	4623 (13.0)
Secondary	15,040 (43.5)	476 (46.0)	15,516 (43.6)
Post-secondary	14,996 (43.4)	435 (42.1)	15,431 (43.4)
Household income (SEK/year)			
<200,000	7598 (22.0)	228 (22.1)	7826 (22.0)
200,000–300,000	8468 (24.5)	277 (26.8)	8745 (24.6)
300,000–400,000	9712 (28.1)	284 (27.5)	9996 (28.1)
>400,000	8758 (25.4)	245 (23.7)	9003 (25.3)
Maternal country of birth			
Nordic country	25,433 (73.6)	815 (78.8)	26,248 (73.8)
Other country	9103 (26.4)	219 (21.2)	9322 (26.2)
Fetal sex			
Male	17,667 (51.2)	544 (52.6)	18,211 (51.2)
Female	16,869 (48.8)	490 (47.4)	17,359 (48.8)
Year of birth			
2000	2858 (8.3)	67 (6.5)	2925 (8.2)
2001	3172 (9.2)	93 (9.0)	3265 (9.2)
2002	3432 (9.9)	101 (9.8)	3533 (9.9)
2003	3723 (10.8)	76 (7.4)	3799 (10.7)
2004	3742 (10.8)	100 (9.7)	3842 (10.8)
2005	3583 (10.4)	117 (11.3)	3700 (10.4)
2006	3491 (10.1)	144 (13.9)	3635 (10.2)
2007	3937 (11.4)	120 (11.6)	4057 (11.4)
2008	3903 (11.3)	123 (11.9)	4026 (11.3)
2009	2695 (7.8)	93 (9.0)	2788 (7.8)
Season of birth			
Winter	7921 (22.9)	233 (22.5)	8154 (22.9)
Spring	9115 (26.4)	319 (30.9)	9434 (26.5)
Summer	9082 (26.3)	230 (22.2)	9312 (26.2)
Autumn	8418 (24.4)	252 (24.4)	8670 (24.4)

**Table 2 ijerph-17-01744-t002:** Summary of modeled black carbon, local PM_2.5_ and PM_10_, total PM_2.5_ and PM_10_, and NO_x_ exposure levels (µg/m^3^) during the entire pregnancy exposure window, over the 2000–2009 period.

Pollutant	*n*	Mean (SD)	Range
Black carbon	32,341	0.36 (0.17)	0.03–1.93
Local PM_2.5_	30,892	1.56 (0.73)	0.13–7.52
Local PM_10_	31,033	2.92 (1.55)	0.20–9.97
Total PM_2.5_	25,050	11.09 (1.16)	6.85–17.30
Total PM_10_	31,039	15.81 (2.35)	10.25–25.62
NO_x_	33,074	14.82 (7.62)	1.01–47.68

**Table 3 ijerph-17-01744-t003:** Adjusted complete case analysis (CCA) for total preeclampsia in relation to the quartiles of all pollutants during each window of exposure.

Pollutant	Exposure Window	Quartile 2 AOR ^§^ (95% CI)	Quartile 3 AOR ^§^ (95% CI)	Quartile 4 AOR^§^ (95% CI)
Black carbon				
	Entire pregnancy	1.17 (0.96, 1.44)	1.32 ** (1.08, 1.62)	1.42 ** (1.16, 1.73)
	1st trimester	1.05 (0.86, 1.28)	1.29 * (1.06, 1.57)	1.33 ** (1.09, 1.63)
	2nd trimester	1.19 (0.98, 1.45)	1.16 (0.95, 1.43)	1.50 ** (1.23, 1.82)
	3rd trimester	1.08 (0.88, 1.31)	1.20 (0.99, 1.46)	1.35 ** (1.11, 1.63)
Local PM_2.5_				
	Entire pregnancy	1.14 (0.93, 1.41)	1.25 * (1.01, 1.53)	1.50 ** (1.22, 1.85)
	1st trimester	1.03 (0.85, 1.26)	1.21 * (1.00, 1.47)	1.34 ** (1.09, 1.64)
	2nd trimester	0.96 (0.79, 1.17)	1.04 (0.85, 1.27)	1.28 * (1.05, 1.57)
	3rd trimester	1.27 * (1.05, 1.55)	1.32 ** (1.08, 1.61)	1.36 ** (1.10, 1.67)
Local PM_10_				
	Entire pregnancy	1.11 (0.90, 1.36)	1.21 (0.98, 1.48)	1.39 ** (1.13, 1.71)
	1st trimester	1.02 (0.84, 1.24)	1.19 (0.99, 1.45)	1.32 ** (1.07, 1.62)
	2nd trimester	1.05 (0.86, 1.28)	1.07 (0.87, 1.30)	1.32 ** (1.07, 1.62)
	3rd trimester	1.22* (1.00, 1.48)	1.24 * (1.01, 1.51)	1.43 ** (1.16, 1.76)
Total PM_2.5_				
	Entire pregnancy	1.05 (0.83, 1.32)	1.25 (0.99, 1.57)	1.40 * (1.08, 1.81)
	1st trimester	1.23 * (1.01, 1.51)	1.17 (0.95, 1.45)	1.24 (0.99, 1.55)
	2nd trimester	1.07 (0.86, 1.34)	1.11 (0.88, 1.41)	1.14 (0.90, 1.45)
	3rd trimester	1.03 (0.83, 1.27)	1.01 (0.81, 1.27)	1.27 * (1.02, 1.59)
Total PM_10_				
	Entire pregnancy	1.02 (0.83, 1.25)	1.15 (0.93, 1.43)	1.43 ** (1.12, 1.82)
	1st trimester	1.20 (0.99, 1.47)	1.33 ** (1.09, 1.63)	1.24 * (1.00, 1.54)
	2nd trimester	1.11 (0.91, 1.35)	1.15 (0.94, 1.40)	1.27 * (1.02, 1.57)
	3rd trimester	1.02 (0.84, 1.24)	1.07 (0.87, 1.31)	1.19 (0.96, 1.48)
NO_x_				
	Entire pregnancy	1.26 * (1.04, 1.53)	1.36 ** (1.12, 1.66)	1.61 ** (1.32, 1.97)
	1st trimester	1.11 (0.91, 1.34)	1.27 * (1.05, 1.54)	1.46 ** (1.20, 1.78)
	2nd trimester	1.15 (0.95, 1.39)	1.31 ** (1.09, 1.59)	1.47 ** (1.21, 1.80)
	3rd trimester	1.35 ** (1.12, 1.63)	1.23 * (1.02, 1.50)	1.51 ** (1.24, 1.84)

^§^ Adjusted for maternal age, body mass index, parity, smoking, diabetes mellitus, gestational diabetes, essential hypertension, gestational hypertension, maternal country of birth, education level, annual household income, fetal sex, and year and season of birth. * *p*-value <0.05 as compared to the lowest quartile. ** *p*-value < 0.01 as compared to the lowest quartile.

**Table 4 ijerph-17-01744-t004:** Adjusted complete case analysis (CCA) for total preeclampsia in relation to the quartiles of all pollutants during each window of exposure (without the intermediate variables ^a^).

Pollutant	Exposure Window	Quartile 2 AOR ^§^ (95% CI)	Quartile 3 AOR ^§^ (95% CI)	Quartile 4 AOR ^§^ (95% CI)
Black carbon				
	Entire pregnancy	1.14 (0.94, 1.39)	1.26 * (1.03, 1.54)	1.37 ** (1.12, 1.67)
	1st trimester	1.03 (0.85, 1.25)	1.27 * (1.05, 1.53)	1.30 * (1.06, 1.58)
	2nd trimester	1.16 (0.96, 1.40)	1.14 (0.93, 1.39)	1.45 ** (1.20, 1.76)
	3rd trimester	1.05 (0.87, 1.28)	1.15 (0.94, 1.39)	1.32 ** (1.09, 1.60)
Local PM_2.5_				
	Entire pregnancy	1.12 (0.91, 1.38)	1.23 * (1.00, 1.51)	1.49 ** (1.22, 1.83)
	1st trimester	1.01 (0.84, 1.23)	1.20 * (0.99, 1.46)	1.32 ** (1.08, 1.62)
	2nd trimester	0.94 (0.77, 1.14)	1.03 (0.85, 1.26)	1.27 * (1.04, 1.55)
	3rd trimester	1.24 * (1.02, 1.50)	1.29 ** (1.06, 1.57)	1.32 ** (1.08, 1.62)
Local PM_10_				
	Entire pregnancy	1.08 (0.88, 1.31)	1.18 (0.96, 1.44)	1.36 ** (1.11, 1.67)
	1st trimester	1.02 (0.85, 1.24)	1.20 (0.99, 1.45)	1.31 ** (1.07, 1.61)
	2nd trimester	1.02 (0.84, 1.24)	1.06 (0.87, 1.29)	1.29 * (1.05, 1.58)
	3rd trimester	1.21 (1.00, 1.47)	1.22 (1.00, 1.48)	1.40 ** (1.14, 1.73)
Total PM_2.5_				
	Entire pregnancy	1.03 (0.82, 1.30)	1.27 * (1.01, 1.59)	1.43 ** (1.11, 1.83)
	1st trimester	1.22 (1.00, 1.48)	1.17 (0.95, 1.44)	1.23 (0.99, 1.53)
	2nd trimester	1.09 (0.88, 1.36)	1.14 (0.91, 1.44)	1.17 (0.92, 1.48)
	3rd trimester	1.01 (0.81, 1.25)	1.02 (0.82, 1.27)	1.26 * (1.01, 1.57)
Total PM_10_				
	Entire pregnancy	1.00 (0.81, 1.22)	1.14 (0.92, 1.40)	1.43 ** (1.12, 1.81)
	1st trimester	1.18 (0.97, 1.44)	1.32 ** (1.08, 1.61)	1.23 (0.99, 1.52)
	2nd trimester	1.09 (0.90, 1.33)	1.15 (0.94, 1.40)	1.28 * (1.04, 1.59)
	3rd trimester	1.00 (0.82, 1.21)	1.06 (0.87, 1.30)	1.19 (0.97, 1.47)
NO_x_				
	Entire pregnancy	1.23 * (1.01, 1.49)	1.31 ** (1.08, 1.59)	1.58 ** (1.29, 1.92)
	1st trimester	1.09 (0.90, 1.32)	1.26 * (1.05, 1.53)	1.42 ** (1.17, 1.72)
	2nd trimester	1.12 (0.93, 1.35)	1.28 ** (1.06, 1.55)	1.45 ** (1.19, 1.76)
	3rd trimester	1.34 ** (1.12, 1.61)	1.22 * (1.01, 1.48)	1.48 ** (1.22, 1.79)

^a^ Intermediate variables are gestational diabetes, gestational hypertension, and essential hypertension. ^§^ Adjusted for maternal age, body mass index, parity, smoking, diabetes mellitus, maternal country of birth, education level, annual household income, fetal sex, and year and season of birth. * *p*-value < 0.05 as compared to the lowest quartile. ** *p*-value < 0.01 as compared to the lowest quartile.

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
