# Peer review of "Maternal Exposure to Ambient Air Pollution and Risk of Preeclampsia: A Population-Based Cohort Study in Scania, Sweden"

_ijerph, 2020, doi:10.3390/ijerph17051744_

Round 1
Reviewer 1 Report
The authors present a well-written analysis of the association between several measures of ambient air pollution and the risk of pre-eclampsia. Strengths of this analysis include: the large sample size, use of the air pollution models to estimate exposure during gestation, and the use of registry data to identify cases of preeclampsia. Several previous studies have also suggested a relationship between higher air pollution exposure during pregnancy and risk of preeclampsia. The authors present a reasonable interpretation of their results given the study limitations. With some moderate revisions, this analysis will contribute to the existing literature.
Major Concerns:
1) The authors selected potential confounders based on a priori knowledge and a DAG, which is an ideal and appropriate method. However, based on their DAG (Figure S1) it appears that gestational diabetes, gestational hypertension, and chronic hypertension are on the causal pathway between air pollution and preeclampsia. Based on the current scientific literature it is feasible that these conditions could be causal intermediates. Therefore, these variables should not be included as covariates in regression models evaluating the association between air pollution and preeclampsia. I recommend including results from the main logistic regression models without these covariates, at the very least as a sensitivity analysis.
2) Given the low levels of BC and PM exposure, I recommend presenting effect estimates in a manner more relevant to the study population. A difference of 1 ug/m3 in BC and 5 ug/m3 PM seems to be much greater than the SD or IQR of the exposure distributions for this study population. For example, in this study it appears very few participants are even exposed to 1 ug/m3 of BC, and presenting linear effect estimates for this large of a difference inflates some of the linear effect estimates to be greater than estimates for the highest quartiles of BC exposure.
Minor
3) Some details regarding the air pollution exposure estimates were missing or hard to follow in methods section. It would be helpful if the authors could address the following:
On Line 333, the text highlighting the strengths of the study indicates that exposure was estimated at women’s home residence. However, these details are not clear in the exposure assessment methods section and I suggest adding some clarification. Also, please indicate if different home addresses during each trimester were used to estimate exposure for women who moved during pregnancy. I suggest including the section which describes the exposure models (currently line 150) before the section outlining the specific exposure variables used in this analysis (line 111) .
4) In Tables B2 and B3 with the air pollution exposure quartiles, please add the range in concentrations for each quartile. For example, in Table B2 were the concentrations of each quartile created to be consistent for both PE with SGA and without SGA analyses?
5) I believe Line 194 should indicate “multivariable analyses” instead of “multivariate analyses”.
Author Response
Response to the reviewers
Thank you, reviewers, for taking the time to review this manuscript and provide valuable input.
Reviewer 1:
Major Concerns:
1) The authors selected potential confounders based on a priori knowledge and a DAG, which is an ideal and appropriate method. However, based on their DAG (Figure S1) it appears that gestational diabetes, gestational hypertension, and chronic hypertension are on the causal pathway between air pollution and preeclampsia. Based on the current scientific literature it is feasible that these conditions could be causal intermediates. Therefore, these variables should not be included as covariates in regression models evaluating the association between air pollution and preeclampsia. I recommend including results from the main logistic regression models without these covariates, at the very least as a sensitivity analysis.
Thank you for pointing this out. We have run the analyses without the intermediates. It did not change the results substantially by including them or not as can be illustrated by comparing Table 3 and 4.
The results of the main logistic regression analyses without the intermediate variables were included in Line 265 of revised manuscript and in Line 39 of revised supplementary materials respectively.
The following text was also added:
Line 205: In addition, the gestational diabetes, gestational hypertension and essential hypertension were identified as intermediate variables based on the DAG. Therefore, we conducted our main analysis with or without these intermediate variables.
Line 260: Without adjusting for gestational diabetes, gestational hypertension and chronic hypertension (variables that could lie in the causal pathway between air pollution exposure and preeclampsia), results did not substantially change (Table 4).
Line 277: Results did not change substantially if not adjusting for the following variables that could be in the causal pathway: gestational diabetes, gestational hypertension, and chronic hypertension as illustrated in Table S5.
2) Given the low levels of BC and PM exposure, I recommend presenting effect estimates in a manner more relevant to the study population. A difference of 1 ug/m3 in BC and 5 ug/m3 PM seems to be much greater than the SD or IQR of the exposure distributions for this study population. For example, in this study it appears very few participants are even exposed to 1 ug/m3 of BC, and presenting linear effect estimates for this large of a difference inflates some of the linear effect estimates to be greater than estimates for the highest quartiles of BC exposure.
We agree with the reviewers and have now changed for the IQR to be main analyses. We have moved the tables of linear exposure to Supplemental material and replaced with the results from IQR. We will, however, still include the linear exposure analysis with these increments as these results could contribute to meta-analyses. The increments used are those recommended and most used for this purpose. To make it more transparent we have added some text:
We have added a sentence in result section stating:
Line 273: It should be stated that the number of women exposed to levels above 1 µg/m3 of BC are low counting 288 (0.6%) cases of CCA.
Further in discussion:
Line 375: Moreover, when interpreting our results from the linear trends it should be considered that few women are exposed to levels above the increment, such as 1µg/m3 of BC.
Minor
3) Some details regarding the air pollution exposure estimates were missing or hard to follow in methods section. It would be helpful if the authors could address the following:
On Line 333, the text highlighting the strengths of the study indicates that exposure was estimated at women’s home residence. However, these details are not clear in the exposure assessment methods section and I suggest adding some clarification. Also, please indicate if different home addresses during each trimester were used to estimate exposure for women who moved during pregnancy. I suggest including the section which describes the exposure models (currently line 150) before the section outlining the specific exposure variables used in this analysis (line 111).
“Exposure assessment” section has been moved to Line 101 before the section of exposure variables. We have also added the following text for clarification:
Line 153: We obtained the geographical coordinates of the residential addresses of each woman from Statistics Sweden and used this data to calculate her individual exposure. As this address database is only updated annually, any changes of residency can only be updated at the end of the calendar year. Therefore, we estimated exposure for every gestational month based on the nearest available time: January-June coordinates from the end of the previous year and July-December coordinates from the end of the year.
4) In Tables B2 and B3 with the air pollution exposure quartiles, please add the range in concentrations for each quartile. For example, in Table B2 were the concentrations of each quartile created to be consistent for both PE with SGA and without SGA analyses?
The range in concentrations for each quartile have been added in Tables B2, B3 and B4. These tables have been moved to the supplementary materials as suggested by the Reviewer 2 and renamed as Table S8, S9, and S10. Indeed, the concentrations of each quartile are consistent for both PE with SGA and PE without SGA analyses.
5) I believe Line 194 should indicate “multivariable analyses” instead of “multivariate analyses”.
The term “multivariate analyses” has been changed to “multivariable analyses” as suggested in Line 197.
The term “whole” has been changed to “entire” in Line 27 and 396.
Reviewer 2 Report
Eclampsia in pregnant women is a severe obstetric condition that threatens both the mother and the fetus. Knowledge concerning the etiopathogenesis of this disease is still fragmentary. Searching for associations with potential risk factors, especially environmental factors, the occurrence of this condition is an essential issue from the public health perspective. I would like to congratulate the authors on their work regarding the subject of eclampsia in the context of air pollution impact on the risk of its occurrence. In my opinion, the overall assessment of the manuscript is very positive, and I recommend to accept the paper for publication in the journal. However, from the reviewer's perspective, I recommend making some small modifications to the text:
- Figure 2 requires editing so that it is legible to the recipient. In the version of the manuscript received for review, the figure is illegible.
- The tables format should adhere to editorial requirements. Besides, in my opinion, authors should consider editing most of the tables.
- Additionally, the additional materials placed at the end of the manuscript should be included as a part of the supplement
Author Response
Response to the reviewers
Thank you, reviewers, for taking the time to review this manuscript and provide valuable input.
Reviewer 2:
1. Figure 2 requires editing so that it is legible to the recipient. In the version of the manuscript received for review, the figure is illegible.
Thank you for pointing this out. The Figure 2 has been changed to a legible figure as requested after Line 98.
2. The tables format should adhere to editorial requirements. Besides, in my opinion, authors should consider editing most of the tables.
We have now moved Appendix A and B to Supplemental material. To our best knowledge, we now adhere to editorial requirements below; please give us some details if there is anything else that needs change.
- All Figures, Schemes and Tables should be inserted into the main text close to their first citation and must be numbered following their number of appearance (Figure 1, Scheme I, Figure 2, Scheme II, Table 1, ).
- All Figures, Schemes and Tables should have a short explanatory title and caption.
- All table columns should have an explanatory heading. To facilitate the copy-editing of larger tables, smaller fonts may be used, but no less than 8 pt. in size. Authors should use the Table option of Microsoft Word to create tables.
3. Additionally, the additional materials placed at the end of the manuscript should be included as a part of the supplement.
We have now moved Appendix A and B to Supplemental material.